# Prevalence of viral hepatitis B in Ghana between 2015 and 2019: A systematic review and meta-analysis

**Julius Abesig**, **Yancong Chen, Huan Wang, Faustin Mwekele Sompo, Irene X. Y. Wu***

Xiangya School of Public Health, Central South University, Changsha, China

* irenexywu@csu.edu.cn

## Abstract

Chronic hepatitis B (HBV) remains a significant public health problem in Ghana and past reviews conducted could not calculate a nationwide prevalence of the disease due to lack of primary research for some regions of the country. We therefore conducted this study to summarize and update the available information on HBV infection burden (prevalence) in Ghana from 2015–2019.We systematically searched PubMed, Embase, ScienceDirect, and Google Scholar to retrieve primary studies published in peer-reviewed journals from November 2015 to September 2019, assessing the prevalence of HBV among the Ghanaian populace. The review included 21 studies across all ten old regions of Ghana with a total sample population of 29 061. The HBV prevalence was estimated for subpopulations as follows: 8.36% in the adult population, 14.30% in the adolescent population, and 0.55% in children under five years (pre-school). Among adults, HBV infection prevalence was the highest in the special occupation group (14.40%) and the lowest prevalence rate of 7.17% was recorded among blood donors. Prevalence was lower in the north than in the southern part of the country. The Ashanti region had the most studies at 6/21 (29%), while no study was identified for the Upper West region. Across the country, the highest HBV infection prevalence rates were recorded in the age group of 20–40 years. The burden of hepatitis B is enormous and remains an important public health issue in Ghana. Addressing the issue will require an integrated public health strategy and rethinking of the implementation gaps in the current HBV infection control program. This will help propel the country towards eliminating the disease by 2030.

## Introduction

Hepatitis B virus (HBV) infection can lead to severe disease and death, affecting people world-wide. About 2 billion people worldwide are estimated to have been exposed to HBV, with almost one quarter of them having a chronic infection [1, 2]. Every year, more than half a million HBV-related deaths are recorded all over the world [1, 3]. Reports have often shown disparities in the levels of endemicity of HBV across the world, with Sub-Saharan Africa (SSA) and East Asia among the high-endemic areas where about 5% and 10% of the adult populace,

**Data Availability Statement:** All relevant data are within the manuscript and its Supporting Information files.

**Funding:** The National Nature Science Foundation of China (No. 81973709): The Hunan Nature Science Foundation (No. 2019JJ40348).

**Competing interests:** The authors have declared that no competing interests exist.

respectively, is chronically infected [3–8]. In Europe and America, about 1% of the population is chronically infected. The risk of being infected with HBV in one's lifetime in most countries in Africa and Asia, including parts of the Middle East, is estimated to be more than 60% [5, 9, 10]. Under-reporting of HBV in Africa makes it difficult to accurately estimate the disease burden, but some estimates suggest that 70–90% of adults showed some evidence of HBV infection and the HBsAg positivity rate is placed at 6–20% [1, 2, 9].

Individuals with chronic HBV infection have an increased risk of liver disease and hepatocellular carcinoma (HCC). It is estimated that 10 to 33% of all individuals who develop a persistent infection will end up with chronic hepatitis, and among them, 20 to 50% are likely to develop liver cirrhosis [11]. HCC is a dangerous cancer with few treatment options that is often a challenge in many third world settings such as Africa [1, 12]. SSA is shown to have one of the highest HBV-related liver cancer rates worldwide [13]. HBV-related liver cancer is more prominent among males than females in the African region [5, 13, 14]. It is also important to mention that the mean age for developing HCC in Africa and other developing regions is younger than what is seen in developed regions such as North America and Western Europe. Therefore, productive years are lost to HCC as a result of HBV infection in SSA [9, 15, 16].

HBV is a disease of public health importance in Ghana and needs the utmost attention. Ghana is located in SSA, a region noted for high ($\geq$8.00%) prevalence of chronic HBV infection compared with the rest of the world [1, 9]. Some recent reviews have estimated and reported the prevalence of HBV in Ghana [17–19]. For instance, a worldwide estimation of HBV prevalence by Schweitzer et al. in 2013 revealed a 12.92% [2] prevalence of chronic HBV infection in Ghana derived from 12 studies. Ofori-Asenso and Agyeman reported a 12.3% prevalence derived from 30 studies published before 2015 [20]. Neither review covered original studies from all regions in Ghana.

Some other studies have also revealed a varying prevalence rate of HBV in Ghana, from 10 to 15% [2, 20]. A search of the literature revealed that no review specifically summarized data on HBV prevalence in Ghana, as Ofori-Asenso and Agyeman reported a limitation that original research was unavailable for four regions and hence based their reported prevalence on six of the ten regions. This reveals that even though there has been a significant effort to estimate the burden of HBV in Ghana, gaps remain in the available evidence. In 2015, the World Health Organization (WHO) set a target to eliminate hepatitis B by the year 2030 [3]. To explore and address any gaps in the available evidence and track the HBV prevalence in Ghana following this declaration [3], we conducted a systematic review to compile updated information on the prevalence of HBV in Ghana from November 2015 –September 2019.

## Methods

### Search strategies

The current systematic review was reported according to the Preferred Reporting Items for Systematic Reviews and Meta-Analysis (PRISMA) [21]. We searched published literature from PUBMED, EMBASE, Google Scholar, and ScienceDirect. The search was conducted using the following keywords: hepatitis B, hepatitis B surface antigen, hepatitis B virus, hepatitis B e antigen, hepatitis B core antigen chronic hepatitis B, HBV, HepB, HBsAg, HBeAg, prevalence, Ghana, and Ghanaian (S1 File.). The search terms were used separately and in combination using the Boolean operators "AND" or "OR" as appropriate. The search ranged from November 1st, 2015 to September 30th, 2019. No language limitations were set during the literature search; however, the search was limited to research on humans.

## Inclusion criteria

The inclusion criteria were as follows: (i) studies exclusively done for populations in Ghana; (ii) articles published in peer-reviewed journals; (iii) cross-sectional studies reporting the prevalence of HBV; (iv) study participants from any subpopulation except studies among patients with HBV; and (v) studies focusing on prevalence using the presence of hepatitis B surface antigen (HBsAg) as a biomarker for diagnosis. We used no language restriction. However, we excluded studies that reported other forms of hepatitis than hepatitis B, book chapters, and case reports.

## Data extraction

Two investigators (JA and HW) independently screened the studies according to the titles and abstracts. If the articles met the eligibility criteria, we further read the full text to screen the study. Any discrepancies between investigators were planned to be resolved by a third investigator (YC). However, no such situation arose.

Using a predesigned data extraction form, JA and HW independently extracted the following data from each study: study characteristics, including the name of the primary author, publication year, study design, sample size, methods of testing HBV, and prevalence of HBV. Any discrepancies were resolved by referring to the original publications.

## Methodological quality assessment

The Newcastle-Ottawa scale (NOS) for cross-sectional studies quality assessment tool was adopted and used to assess the quality of each study [22]. The tool has three major sections. The first section, graded out of five stars, focuses on the sample selection of each study. The second section deals with the comparability of the study, and the last section deals with the outcomes and statistical analysis of each original study. A total NOS score ranges from 0 to 10. Study scores of 9–10 points were considered very good, 7–8 points as good, 5–6 points as satisfactory and <5 as unsatisfactory (S2 File) [22]. Two authors (JA and HW) independently assessed the quality of each original study using the tool. Disagreements between the two authors were resolved through discussion.

## Data analysis

HBV infections were measured with prevalence and 95% confidence interval (CI). The review considered the regional distribution of studies based on the ten old regions and combined the six recently established regions to their mother regions. Thus, Greater Accra, Upper West, Upper East, Ashanti, Eastern, and Central regions stood alone. However, in this study the Northern region consists of Northeast, Savannah, and Northern regions; Brong-Ahafo consists of Bono, Bono East, and Ahafo regions; and the Western region consists of Western and Western North regions.

Meta-analysis with a random effect model was used to pool the prevalence from different homogeneous studies. Heterogeneity was tested using the Q test with p<0.1 considered as statistical heterogeneity. The degree of heterogeneity was measured with $I^2$. Subgroup analysis was done if the presence of substantial heterogeneity was observed ($I^2 > 50\%$). Due to the heterogeneity among the included studies, meta-analyses were conducted separately according to different populations (pre-school children, adolescents, and adults). A similar rule was applied to the adult population, among whom meta-analyses were conducted according to subpopulations with different characteristics (e.g. blood donors, patients with HIV). A fixed effect cumulative analysis was conducted beginning with the largest sample size and adding subsequent

studies in the same order while estimating the relative weight to explore the possible impact of bias from smaller studies on the overall effect size. The point at which any additional study did not show any significant shift on the effect size was considered the cut-off point and the conclusion was then made. Statistical significance was always set at p < 0.05 except for in the heterogeneity tests. Subgroup analyses (e.g. northern and southern Ghana, urban and rural) were done within each subpopulation when statistical heterogeneity was found. All the meta-analyses were performed with Comprehensive Meta-analysis Software for Academics (standard v. 3).

### Ethical approval

The study did not require ethical approval as it used data retrieved from published studies that are already in the public domain.

## Results

### Results of study identification and retrieval

Fig 1 records the process and detailed results of the literature search and selection. A total of 150 articles were retrieved from the databases and additional sources in the literature search. Thirty-eight articles were retained for full-text review after the removal of duplicates and ineligible studies based on the titles and abstracts. Of the 38 studies, 21 articles met the inclusion criteria for the systematic review and 19 were finally added to the meta-analysis.

### Basic characteristics of included studies

The studies (Table 1) included in the review had total sample sizes of 29 061 and 28 724 respectively for the systematic review and meta-analysis, respectively, across the country. The largest number of studies (6/21, 29%) was conducted in the Ashanti region while no study was identified in the Upper West region. Table 1 shows the details of all studies included in the review, covering nine out of the ten old regions. The majority of the retained studies were conducted for the adult population (18/21, 86%) and urban settings (13/21, 62%). No study was conducted in a male-only population, but 6/21 (29%) were conducted in a female-only population (pregnant women), 4/21 (19%) among patients with HIV, and 3/21 studies (19%) in the blood donor population. Regarding the methodological quality of the included studies, 14%, 76%, and 10% of studies were judged as very good, good, and satisfactory quality, respectively, based on the NOS score, while no study was assessed as unsatisfactory.

### Overall prevalence of hepatitis B infection

The reported prevalence among the 21 studies in the systematic review ranged from 0.05% to 54.20% (Table 1). The prevalence among the 19 studies included for the meta-analysis ranged from 0.05% to 16.70% (Table 1). Generally, there was diversity in participants' ages; however, about 85.71% of the included studies were conducted in adult populations (>20 years). Two studies were conducted in children less than five years. The most prevalent age group ranged from 20 to 40 years, as specified in ten studies [17, 19, 26, 29, 34–36, 38, 40, 41]. There was no overall pooled national HBV prevalence for the included studies in the meta-analysis because of the vast differences among study participants.

### Prevalence among pre-school children

Two studies (528 participants in total) reported HBV infection prevalence among pre-school children, at 0.05% and 1.90%. The pooled prevalence was 0.55% (95% CI: 0.02–14.23%) (S1

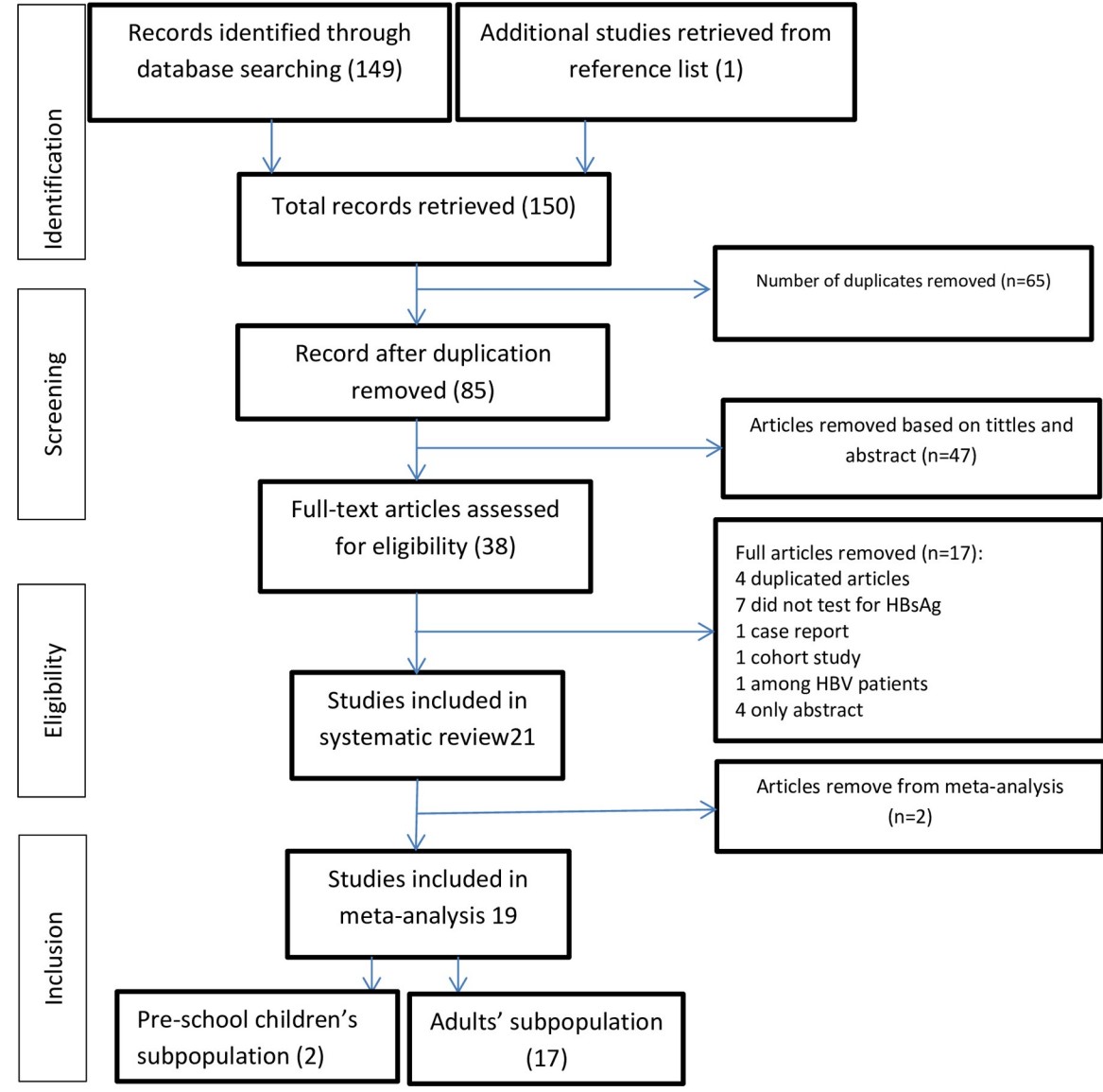

**Fig 1. Flow chart of study search and retrieval process (adopted PRISMA: 2009).**

Appendix). There was moderate heterogeneity ($I^2$ = 60.84%, p = 0.110) between these two studies.

## Prevalence among adolescents

Only one study was conducted in an adolescent population, with 182 participants from a senior high school in a rural setting. The HBV prevalence was 14.30% (95% CI: 9.20–19.37%) (Table 1).

## Prevalence among adults

Seventeen studies were considered for the meta-analysis in the adult population with a total sample size of 28 196. The HBV prevalence ranged from 2.45% in the Northern region to

**Table 1. Summary of the descriptive characteristics of articles included in the study.**

| Authors' details | Study region | Study population | Age group (mean/range) | Site | Sample size | Test | HBsAg+(%) | Quality assessment |
|---|---|---|---|---|---|---|---|---|
| **Pre-school children** | | | | | | | | |
| Dassah, 2015 [23] | Upper East | Vaccinated children | 3.1 yrs | Rural | 104 | ELISA | 1.90 | Good |
| Apiung, 2017 [24] | Greater Accra | Vaccinated children | 1.3 yrs | Urban | 424 | ELISA | 0.05 | Satisfactory |
| **Adolescents** | | | | | | | | |
| Dzidzinyo, 2016 [25] | Volta | School children | 17.4 yrs | Rural | 182 | ELISA | 14.30 | Good |
| **Adults** | | | | | | | | |
| Adoba, 2015 [26] | Ashanti | Barbers | 28.2 yrs | Urban | 200 | RDT | 14.50 | Good |
| Ephraim, 2015 [27] | Ashanti | Pregnant women | 27.0 yrs | Rural | 168 | RDT | 9.50 | Good |
| Ampah, 2016 [28] | Ashanti | Community members | 25.3 yrs | Rural | 1323 | RDT | 8.10 | Very good |
| Adjei, 2016 [29] | Greater Accra | Long-distance drivers | 40.6 yrs | Urban | 106 | ELISA | 14.20 | Satisfactory |
| Archampong, 2016 [30] | Greater Accra | Patients with HIV | 41.0 yrs | Urban | 3108 | ELISA | 8.30 | Good |
| Kye-Duodo, 2016 [17] | Eastern | Patients with HIV | 40.0 yrs | Mixed | 320 | ELISA | 8.80 | Good |
| Luuse, 2016 [31] | Volta | Pregnant women | 27.7 yrs | Urban | 208 | RDT | 2.40 | Good |
| Lokpo, 2017 [32] | Eastern | Blood donors | 20–50 yrs | Urban | 11 436 | RDT | 7.20 | Very good |
| Lokpo, 2017 [33] | Volta | Blood donors | 18–58 yrs | Urban | 4180 | RDT | 6.90 | Very good |
| Volker, 2017 [19] | Western | Pregnant women | 26.3 yrs | Rural | 174 | RDT | 16.70 | Good |
| Osei, 2017 [34] | Volta | Blood donors | 20.0–40.0 yrs | Mixed | 576 | RDT | 7.50 | Good |
| Helegbe, 2018 [35] | Northern | Pregnant women | 28.5 yrs | Urban | 3127 | RDT | 4.20 | Good |
| Owusu, 2018[36] | Ashanti | Patients with jaundice | 35.0 yrs | Urban | 155 | PCR & RDT | 54.20 | Good |
| Anabire, 2019[37] | Northern | Pregnant women | 27.8 yrs | Urban | 2071 | RDT | 7.70 | Good |
| Boateng, 2019 [38] | Ashanti | Patients with HIV | 40.9 yrs | Urban | 400 | RDT | 12.50 | Good |
| De Mendoza, 2019 [39] | Ashanti | Outpatients | 26.0 yrs | Urban | 305 | RDT | 8.50 | Good |
| Frempong, 2019 [40] | Brong-Ahafo | Pregnant women | 28.9 yrs | Mixed | 100 | ELISA | 10.00 | Good |
| Pappoe, 2019 [41] | Central | Patients with HIV | 41.0 yrs | Urban | 394 | RDT | 6.60 | Good |

ELISA, enzyme-linked immunoassay; HIV, human immunodeficiency virus; PCR, polymerase chain reaction; RDT, rapid diagnostic test.

16.70% in the Western region. The pooled prevalence was 8.36% (CI: 7.30–9.60%) (Fig 2) with a high level of heterogeneity ($I^2 = 86.52\%$, p<0.001). Subgroup analyses were conducted, and HBV prevalence was reported according to different subgroups among adult populations. In the subgroup analysis (Table 2), two subgroups (geographic zone and sample size) showed statistically significant differences between groups (p<0.020).

## Prevalence of hepatitis B in the general population (community members and outpatients)

Two studies [28, 39] were conducted in the general population (community members and outpatients) with a total sample size of 1628 participants. The hepatitis B prevalences were 8.10% and 8.50%, with a pooled prevalence of 8.18% (CI: 6.94–9.61%) (S2 Appendix). There was no evidence of heterogeneity ($I^2 = 0\%$, p = 0.818).

## Prevalence of hepatitis B among blood donors

A total of three studies [32–34] reported HBV prevalence among blood donors with a total sample size of 16 192 participants. The hepatitis B prevalence ranged from 6.94% to 7.50% with a pooled prevalence of 7.17% (CI: 6.78–7.57) (S3 Appendix). There was no evidence of heterogeneity ($I^2 = 0\%$, p = 0.784).

# Random effects model

| Study name | | Statistics for each study | | | | | Event rate and 95% CI |
|---|---|---|---|---|---|---|---|
| | | Event rate | Lower limit | Upper limit | Z-Value | p-Value | |
| Lokpo | 2017 | 0.0723 | 0.0677 | 0.0772 | -70.6758 | 0.0000 | |
| Lokpo | 2017 | 0.0694 | 0.0621 | 0.0775 | -42.6525 | 0.0000 | |
| Helegbe | 2018 | 0.0420 | 0.0355 | 0.0496 | -35.0771 | 0.0000 | |
| Archampong | 2016 | 0.0830 | 0.0738 | 0.0932 | -36.9475 | 0.0000 | |
| Anabire | 2019 | 0.0770 | 0.0663 | 0.0893 | -30.1340 | 0.0000 | |
| Ampah | 2016 | 0.0810 | 0.0675 | 0.0970 | -24.1034 | 0.0000 | |
| Osei | 2017 | 0.0750 | 0.0561 | 0.0995 | -15.8813 | 0.0000 | |
| Boateng | 2019 | 0.1250 | 0.0960 | 0.1612 | -12.8710 | 0.0000 | |
| Pappoe | 2019 | 0.0660 | 0.0453 | 0.0952 | -13.0590 | 0.0000 | |
| Kye-Duodo | 2016 | 0.0880 | 0.0615 | 0.1244 | -11.8499 | 0.0000 | |
| De Mendoza | 2019 | 0.0850 | 0.0585 | 0.1220 | -11.5735 | 0.0000 | |
| Luuse | 2016 | 0.0240 | 0.0100 | 0.0564 | -8.1790 | 0.0000 | |
| Adoba | 2015 | 0.1450 | 0.1027 | 0.2009 | -8.8354 | 0.0000 | |
| Volker | 2017 | 0.1670 | 0.1186 | 0.2299 | -7.9065 | 0.0000 | |
| Ephraim | 2015 | 0.0950 | 0.0590 | 0.1495 | -8.5665 | 0.0000 | |
| Adjei | 2016 | 0.1420 | 0.0875 | 0.2221 | -6.4642 | 0.0000 | |
| Frempong | 2019 | 0.1000 | 0.0547 | 0.1760 | -6.5917 | 0.0000 | |
| | | 0.0836 | 0.0728 | 0.0958 | -31.2739 | 0.0000 | |

-0.25   -0.13   0.00   0.13   0.25

**Test of Heterogeneity:[I2=86.52%, p<0.001]**

**Fig 2. Forest plot of hepatitis B infection prevalence rate in the adult population in Ghana from 2015 to 2019.**

## Prevalence of hepatitis B among patients with HIV

The meta-analysis was based on four studies [17, 30, 38, 41] conducted among HIV-positive clients. The total sample size was 4222 participants. The HBV prevalence ranged from 6.60% in the Central region to 12.50% in the Ashanti region. The pooled prevalence was 8.92% (CI: 7.03–11.25, p<0.001) (S4 Appendix). There was moderate heterogeneity among the included studies ($I^2$ = 69.91%, p = 0.019). As a result, a further subgroup analysis (S5 Appendix) was performed to explore the source of heterogeneity. Statistical variations among the included studies were explained by the type of test used in the diagnosis of HBV (ELISA or RDT) among the study participants. There was, however, no statistical difference between the two subgroups (p = 0.732).

## Prevalence of hepatitis B infection among pregnant women

Six studies [19, 27, 31, 35, 37, 40] were conducted among pregnant women with a total sample size of 5848 participants. The prevalence ranged from 2.4% to 16.7% with a pooled prevalence of 7.44% (CI; 4.71–11.55) and p<0.001 (S6 Appendix). The results showed very high heterogeneity ($I^2$ = 92.69%, p<0.001). Further subgroup analysis (S7 Appendix) revealed that the year of the study contributed to the heterogeneity of the results. Two studies were conducted in 2019 with a total sample size of 2171. The pooled prevalence was 7.82% (CI: 6.78–9.03). There was, however, no evidence of heterogeneity among these studies ($I^2$ = 0, p = 0.404).

**Table 2. Analysis of the change in hepatitis B prevalence in different subgroups in the adult population in Ghana (2015–2019).**

| Subgroups | No. of studies (no. of participants) | Pooled prevalence (%) (95% CI) | $I^2$ (%) | P-value (subgroup differences) |
|---|---|---|---|---|
| **HIV status** | | | | 0.616 |
| HIV+ | 4 (4222) | 8.91 (6.71–11.73) | 69.91 | |
| HIV- | 13 (23 974) | 8.19 (6.95–9.64) | 87.75 | |
| **Residency** | | | | 0.273 |
| Rural | 3 (1665) | 10.65 (7.63–14.68) | 84.76 | |
| Urban | 11 (25 532) | 7.84 (6.62–9.27) | 89.35 | |
| Mixed | 3 (996) | 8.47 (5.89–12.04) | 0.00 | |
| **Pregnancy status** | | | | 0.309 |
| Pregnant | 6 (5848) | 8.78 (7.42–10.36) | 75.91 | |
| Not pregnant | 11 (22 348) | 7.49 (5.78–9.65) | 92.70 | |
| **Test type** | | | | 0.303 |
| ELISA | 4 (3634) | 9.68 (7.08–13.11) | 36.70 | |
| RDT | 13 (24 562) | 8.06 (6.85–9.45) | 88.65 | |
| **Sample size** | | | | 0.005** |
| Large (>1000) | 6 (27 045) | 6.94 (5.80–8.29) | 89.84 | |
| Small (<1000) | 11 (1151) | 9.88 (8.31–11.71) | 74.34 | |
| **Blood donor status** | | | | 0.294 |
| Donor | 3 (16 192) | 7.20 (5.19–9.92) | 0.00 | |
| Not donor | 14 (12 004) | 8.75 (7.38–10.35) | 88.52 | |
| **Zones** | | | | 0.018** |
| Northern | 2 (5198) | 5.72 (4.03–8.05) | 96.44 | |
| Southern | 15 (22 998) | 8.93 (7.73–10.30) | 79.65 | |

HIV, human immunodeficiency virus; RDT, rapid diagnostic test; ELISA, enzyme-linked immunoassay; CI, confidence interval.

** P-value of subgroup differences <0.020.

## Prevalence of hepatitis B among special occupational populations (barbers and long-distance drivers)

Two studies [26, 29] with a total sample size of 306 participants were conducted among special occupation populations. The prevalences were 14.20% (barbers) and 14.50% (drivers), with a pooled prevalence of 14.40% (CI: 10.89–18.79) (S8 Appendix). There was no evidence of heterogeneity ($I^2$ = 0%, p = 0.943).

## Prevalence of hepatitis B among patients with jaundice

One study [36] was conducted among a symptomatically jaundiced population with 155 participants from an urban setting in the Ashanti region. This region is the most populous in Ghana. The study was hospital-based and reported an extremely high HBV prevalence of 54.20% among the study population. Participants who tested positive with PCR, RDT, or both were considered positive for hepatitis B infection.

## Cumulative meta-analysis

Bias from smaller studies was explored with cumulative analysis, the results from which indicated that smaller studies did not impact HBV prevalence. Results based on precision (large sample size) showed that 8/17 studies with a relative weight of 91.34% estimated a pooled effect

# Fixed effects model

| Study name | | Cumulative statistics | | | | | Cumulative event rate (95% CI) | Relative weight |
|---|---|---|---|---|---|---|---|---|
| | | Point | Lower limit | Upper limit | Z-Value | p-Value | | |
| Lokpo | 2017 | 0.0723 | 0.0677 | 0.0772 | -70.6758 | 0.0000 | | 40.53 |
| Lokpo | 2017 | 0.0715 | 0.0676 | 0.0757 | -82.5465 | 0.0000 | | 54.79 |
| Helegbe | 2018 | 0.0676 | 0.0641 | 0.0713 | -89.4911 | 0.0000 | | 61.44 |
| Archampong | 2016 | 0.0700 | 0.0667 | 0.0735 | -96.7682 | 0.0000 | | 73.94 |
| Anabire | 2019 | 0.0706 | 0.0674 | 0.0740 | -101.3446 | 0.0000 | | 81.71 |
| Ampah | 2016 | 0.0712 | 0.0681 | 0.0745 | -104.1618 | 0.0000 | | 86.92 |
| Osei | 2017 | 0.0713 | 0.0682 | 0.0745 | -105.3649 | 0.0000 | | 89.03 |
| Boateng | 2019 | 0.0724 | 0.0693 | 0.0756 | -106.0707 | 0.0000 | | 91.34 |
| Pappoe | 2019 | 0.0723 | 0.0692 | 0.0755 | -106.8704 | 0.0000 | | 92.62 |
| Kye-Duodo | 2016 | 0.0725 | 0.0694 | 0.0757 | -107.5200 | 0.0000 | | 93.98 |
| De Mendoza | 2019 | 0.0726 | 0.0696 | 0.0758 | -108.1378 | 0.0000 | | 95.23 |
| Luuse | 2016 | 0.0724 | 0.0694 | 0.0756 | -108.4167 | 0.0000 | | 95.49 |
| Adoba | 2015 | 0.0731 | 0.0701 | 0.0763 | -108.7084 | 0.0000 | | 96.80 |
| Volker | 2017 | 0.0739 | 0.0709 | 0.0771 | -108.9001 | 0.0000 | | 98.08 |
| Ephraim | 2015 | 0.0741 | 0.0710 | 0.0773 | -109.2317 | 0.0000 | | 98.84 |
| Adjei | 2016 | 0.0744 | 0.0714 | 0.0776 | -109.3918 | 0.0000 | | 99.52 |
| Frempong | 2019 | 0.0745 | 0.0715 | 0.0777 | -109.5860 | 0.0000 | | 100.00 |
| | | 0.0745 | 0.0715 | 0.0777 | -109.5860 | 0.0000 | | |

-0.25    -0.13    0.00    0.13    0.25

Test of Heterogeneity:[I2=86.52%, p=0.001]

**Fig 3. Cumulative forest plot for the included studies (fixed effects model) in the adult-only population.**

of 7. 24% with a 95% CI (6.93 to 7.56%) included the pooled prevalence of 7.45% for the total of 17 studies in the meta-analysis (Fig 3).

## Discussion

This study was conducted with the aim of summarizing the updated national prevalence of HBV in Ghana from 2015 to 2019. However, there was no study identified for the Upper West region and thus, the prevalence of HBV from the remaining regions was summarized according to different populations due to the heterogeneity of participants included. A national survey that covers all regions in the country to estimate the actual burden of the disease in Ghana is needed.

The pooled HBV prevalence was very low (0.55%) among pre-school children. These were children who have been vaccinated against HBV with the pentavalent vaccine. Although there was moderate heterogeneity, the variation in age between studies partly explains this. As the mean age increases from 1.3 years to 3.1 years, the antibodies developed through immunization tend to decrease [23, 24] in individuals and thus older children are at a higher risk of infection. This revelation emphasizes the importance of vaccination and good immunization coverage among Ghanaian children [42].

It was noted that Ghana started a national expanded program for hepatitis immunization in 2002 [42], which should have reduced the prevalence of HBV infections, especially among adolescents. However, an alarming prevalence of about 14.30% was identified among adolescents in this study. There were not sufficient studies to explore this; however, adolescence is characterized by youthful exuberance, sexual exploitation, body modifications (tattooing, piercing, etc.), and risky sexual behaviors, which are all risk factors for HBV transmission [43]. Furthermore, in Ghana, adolescents do not discuss sex openly with their parents; they resort to friends who are as well inexperienced with sexually related matters and may end up in engaged in

risky behaviors. This suggests the need for further research to fully explore the cause of the high HBV prevalence in this population, regardless of the national immunization program.

In the adult population, overall HBV prevalence was found to be 8.36% in Ghana based on HBsAg seropositivity. This emphasizes that HBV remains a major public health concern in the country. In contrast, the current prevalence is significantly lower than the prevalence of 12.30% reported by earlier studies [2, 20, 44, 45]. However, a similar prevalence rate of 8.50% was reported quite recently in an epidemiological study in 2012 [46]. The results of this review are lower than the >8.80% reported in other parts of West Africa [47–49] and the prevalence for the African region at large [2]. For instance, a meta-analysis reported that the pooled prevalence of HBV in the Nigerian population is 13.6% [50]. A high prevalence was reported among different subgroups as well: 14.0% for blood donors, 14.1% for pregnant women attending antenatal clinics, 11.5% for children, and 14.0% for adults [50]. Nonetheless, the results from our study confirm the categorization of Ghana as a high HBV-endemic country (≥8.00%) which exceeds the reported global prevalence rate of 3.61% [1, 3, 7]. Comparing to other parts of the world, the HBV prevalence rate has been estimated to be <5.10% in Iraq and 4.00% in Singapore [2]. The Middle East, Europe, and North America all reported a rate lower than 2.00% [2]. This comparison, therefore, indicates the enormous burden of HBV infection in Ghana.

In the blood donor population, HBV prevalence was 7.17%. Transfusion safety of blood and blood products in Ghana should be a major concern as about one in every ten blood donors is infected with HBV. Our result was lower than the 7.70% and 9.70% reported in Burkina Faso [51] and Sierra Leone, respectively [8]. However, a much lower risk of HBV infection through blood transfusion (1.4 in 100 000) was reported in Korea [52]. Our results showed that rapid diagnostic test (RDT) kits were used in all the studies among blood donors. Furthermore, RDTs are the most commonly used kits in developing countries because of their ease of use and lower cost. Meanwhile, Mutocheluh and colleagues examined five brands of these kits on the Ghanaian market, finding a sensitivity lower than 60% and raising serious concerns for their use in blood donation centers to exclude HBV [53]. The present study emphasizes the need for robust screening techniques and uncompromising adherence to the transfusion policy in Ghana which advocates for the need to screen for transfusion-transmissible diseases such as HIV, HBV/HCV, and syphilis.

Among pregnant women, HBV prevalence remains high (7.44%), as reported in other parts of the sub-region [54] and thus, mother to child transmission remains a major route of HBV infection in Ghana [55]. Comprehensive preconception screening, education, and early initiation of treatment for infected couples can complement the already existing policy on prevention of mother to child transmission in reducing infection risk to the child and also benefit the expectant mothers. Ghana as a country should also consider vaccinating all tested-negative pregnant women as part of the already existing free maternal care service program.

Additionally, this study revealed a very high HBV prevalence of 14.40% among barbers and long-distance drivers. A previous study revealed poor knowledge about HBV transmission among barbers in an urban area in Ghana [56] and thus emphasizes the need for education on HBV transmission in this population.

HBV prevalence among patients with HIV was determined to be 8.92%, although this rate is lower than the 13.60% reported in a previous systematic review by Agyeman and Ofori-Asenso in 2016 [18]. This prevalence is higher than the WHO's category for high-endemic areas. Moreso, this result suggests a high double disease burden for the affected clients and may lead to dire consequences of liver disease in the already immune-compromised persons. Although HBV testing is now considered among patients with HIV, there exists a gap in giving complete care to this population [42]. Patients who test negative for HBV are not protected by

way of vaccination and this remains a missed opportunity in reducing the prevalence in the general population.

The WHO's target to eliminate HBV by the year 2030 seems unachievable in Ghana if the trends in the healthcare system are not realigned towards universal care in the face of this high prevalence. Several factors are implicated in the observed high HBV prevalence in Ghana. There still exists an information gap and a lack of understanding among the Ghanaian populace of the transmission dynamics of HBV. Some studies conducted among pregnant women in the Kintampo and Upper West regions reported low awareness and knowledge about HBV transmission [57, 58]. Mkandawire and colleagues pointed out that, in the Upper West region, the populace is even confused about the difference between HBV infection and other diseases like malaria and yellow fever [59].

The most efficient and cost-effective way of controlling HBV is by vaccination [3, 60]. Interestingly, the high cost of the vaccine has led to many miss opportunities in various service points in the health system. For instance, negative individuals miss out on vaccination at blood donor centers, antenatal care services, and antiretroviral therapy clinics simply because they cannot afford it. These individuals later become positive and increase the overall disease burden. Furthermore, the cost of the monovalent HBV vaccine given to infants born to reactive mothers is borne by parents; this is another financial barrier that may put the child at risk of infection. A subsidy to the cost of these vaccines would dramatically reduce infection rates.

## Strengths and limitations

This current review provides the most updated disease burden figures regarding HBV prevalence in Ghana. As 90.00% of the included studies' methodologies were robust, this result for HBV prevalence can be said to reflect the current situation in Ghana. This study provides the prevalence among children under five years of age which was not identified by previous systematic reviews [2, 20] and thus reaffirms the importance of childhood immunization programs.

There are several limitations to this review. Although we comprehensively searched the literature, an overall national prevalence of HBV could not be determined because we were not able to identify any related publications for the Upper West region. The prevalence determined in this study was therefore based on subpopulations and hence cannot be said to fully represent the general Ghanaian population. Furthermore, most of the studies (13/21, 62%) were conducted in urban settings and in adult populations (18/21, 86%). Hence, the prevalence is more representative of the adult, urban population. The aforementioned information also revealed the research priorities for future studies: HBV prevalence among children and adolescents, in populations from the Upper West region of Ghana, and in rural areas. Due to the limited number of studies included in each meta-analysis, we were not able to explore the potential of publication bias [41]. However, a cumulative bias analysis showed that eight (n = 23 094) large included studies contributed about 91.21% of the total prevalence rate and hence the prevalence was not impacted by smaller studies.

## Conclusions

This systematic review and meta-analysis indicated that the prevalence of HBV infection between 2015 and 2019 in Ghana was 0.55%, 14.30%, and 8.36% among pre-school children, adolescents, and adults, respectively. There was a significantly low prevalence among preschool children and a decline in the adult population in Ghana. There however still remains a high prevalence among adolescents. The burden of chronic HBV infection is also unevenly distributed among the various subpopulations in the adult group and remains high per the

WHO's level of categorization. That notwithstanding, HBV eradication is possible by vaccination. To address these challenges of HBV infection among Ghanaians, there is an urgent need for an integrated approach in the health care system continuum. The government and its partner agencies, therefore, need to explore the possibility of a free HBV vaccination policy with the goal of eliminating this disease burden.

## Supporting information

**S1 File. Detailed database search strategies used to retrieve articles.**
(PDF)

**S2 File. Newcastle-Ottawa Scale adapted for cross-sectional studies.**
(PDF)

**S1 Appendix. Forest plot of HBV prevalence among pre-school children in Ghana.**
(PDF)

**S2 Appendix. Forest plot of HBV prevalence among the general population (community members and outpatients) in Ghana.**
(PDF)

**S3 Appendix. Forest plot of HBV prevalence among blood donors in Ghana.**
(PDF)

**S4 Appendix. Forest plot of HBV prevalence among patients with HIV in Ghana.**
(PDF)

**S5 Appendix. Forest plot of subgroup analysis of hepatitis B prevalence among patients with HIV and by test type.**
(PDF)

**S6 Appendix. Forest plot of HBV prevalence among pregnant women in Ghana.**
(PDF)

**S7 Appendix. Forest plot of hepatitis B prevalence subgroup analysis among pregnant women by study publication year.**
(PDF)

**S8 Appendix. Forest plot of HBV prevalence among special occupation populations (barbers and long-distance drivers) in Ghana.**
(PDF)

**S1 Checklist. PRISMA checklist.**
(PDF)

**S1 Dataset. Minimal dataset.**
(XLSX)

## Acknowledgments

We wish to thank the staff of the epidemiology and health statistics department for their technical support. We would like to thank Editage (www.editage.cn) for English language editing.

## Author Contributions

**Conceptualization:** Julius Abesig, Yancong Chen, Huan Wang, Faustin Mwekele Sompo, Irene X. Y. Wu.

**Data curation:** Julius Abesig.

**Formal analysis:** Julius Abesig, Irene X. Y. Wu.

**Funding acquisition:** Irene X. Y. Wu.

**Methodology:** Julius Abesig.

**Writing – original draft:** Julius Abesig, Irene X. Y. Wu.

**Writing – review & editing:** Julius Abesig, Huan Wang, Irene X. Y. Wu.

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
