## [Decision Letter · Decision Letter 0]

9 Apr 2020

PONE-D-20-02736

Prevalence of viral hepatitis B in Ghana between 2015 and 2019: a systematic review and meta-analysis

PLOS ONE

Dear Dr. Abesig,

Thank you for submitting your manuscript to PLOS ONE. After careful consideration, we feel that it has merit but does not fully meet PLOS ONE’s publication criteria as it currently stands. Therefore, we invite you to submit a revised version of the manuscript that addresses the points raised during the review process.

We would appreciate receiving your revised manuscript by May 24 2020 11:59PM. To enhance the reproducibility of your results, we recommend that if applicable you deposit your laboratory protocols in protocols.io, where a protocol can be assigned its own identifier (DOI) such that it can be cited independently in the future. For instructions see: http://journals.plos.org/plosone/s/submission-guidelines#loc-laboratory-protocols

We look forward to receiving your revised manuscript.

Kind regards,

Jason Blackard, PhD

Academic Editor

PLOS ONE

Journal Requirements:

1. Please ensure that you include a title page within your main document. We do appreciate that you have a title page document uploaded as a separate file, however, as per our author guidelines (http://journals.plos.org/plosone/s/submission-guidelines#loc-title-page) we do require this to be part of the manuscript file itself and not uploaded separately.

Additional Editor Comments (if provided):

This is a meta-analysis of HBV infection in Ghana.  Given the high prevalence of HBV in West Africa, this is a reasonable study goal.  The methods are well described and appropriate.  The manuscript should be reviewed carefully by a native English speaker and/or a professional editing service.

It is unclear why only data from 2015 to 2019 were included?  Certainly there are studies before that time period that are relevant.

Line 2:  Approximately 2 billion people have been exposed to HBV, but there certainly are not that many that are currently infected.

It would be helpful to mention and discuss briefly any other meta-analysis of HBV prevalence conducted in other Africa countries – particularly West Africa – for comparison.

Reviewers' comments:

Reviewer's Responses to Questions

**Comments to the Author**

1. Is the manuscript technically sound, and do the data support the conclusions?

Reviewer #1: No

Reviewer #2: Yes

2. Has the statistical analysis been performed appropriately and rigorously? 

Reviewer #1: No

Reviewer #2: Yes

3. Have the authors made all data underlying the findings in their manuscript fully available?

Reviewer #1: Yes

Reviewer #2: Yes

4. Is the manuscript presented in an intelligible fashion and written in standard English?

Reviewer #1: Yes

Reviewer #2: Yes

5. Review Comments to the Author

Reviewer #1: The systematic review and meta analysis done have not added any additional or new information to the prevalence of Hepatitis B in Ghana. This is due to limitations the authors themselves recognized. Studies do not cover the whole population, most studies are small and the number of studies in the heterogeneous populations few over the short period thus the basis for the meta analysis in some cases were two or three studies. The process thus looked more like an academic exercise rather than providing additional scientific knowledge. A previous meta-analysis covered 10 years: Ofori-Asenso R, Agyeman AA. Hepatitis B in Ghana: a systematic review & meta-analysis of prevalence studies (1995-2015). BMC Infect Dis. 2016 Mar 18;16:130. doi: 10.1186/s12879-016-1467-5. PMID: 26987556; PMCID: PMC4797341. Im not sure why the authors choose to do same over five years 2015-2019 instead of fifteen years, 1995 -2019

Results-There was no PRISMA diagram. There are no random effect plots

Discussion- This needed to have more depth. Ghana started a national expanded programme on Hepatitis immunization in

2002. The high prevalence in adolescents in contrast to children and adults to be put in context. Also the quality or generation of HBsAg tests used should have been analyzed in the meta analysis to enrich the discussions.

Conclusion-The study could not achieve its objective of determining a National prevalence. Even though a negative result in itself is not a problem, I believe the meta analysis covers a very narrow period and the many limitations contributed to their inability to determine a National prevalence.

Reviewer #2: ACCEPT with some good grammatical revisions please

This is a well-conducted and useful meta-analysis. The methods in the literature search and statistical approaches are appropriate and sound. The subgroup analyses were very revealing in terms of pockets of high-need populations within Ghana. These were clearly linked to explanations within the discussion that describe attributes which predispose to infection and/or increased morbidity (i.e. adolescent sexual practices; HBV detection kids with poor sensitivity in the blood donor population; etc.).

The pooled prevalence itself is sufficient to raise alarm bells, but the extremely high rates in certain populations (adolescents, pregnant women, HIV+) suggest additional need for targeted intervention and care.

6. PLOS authors have the option to publish the peer review history of their article (what does this mean?). If published, this will include your full peer review and any attached files.

Reviewer #1: No

Reviewer #2: No

---

## [Author Response · Author response to Decision Letter 0]

16 May 2020

Dr. Jason Blackard

Academic Editor

PLOS One

Dear Dr. Blackard,

Thank you very much for giving us the opportunity to review and improve our manuscript titled“Prevalence of viral hepatitis B in Ghana between 2015 and 2019: a systematic review and meta-analysis” (PONE-D-20-02736). We found your comments invaluable for improving our manuscript. We have addressed each comment in a point-by-point format below, and we have also revised the manuscript according to these comments with tracked changes. A clean version of the revised manuscript has also been included.

We would like to express our sincere gratitude for your input on our submission. Please do not hesitate to contact us if further modifications are needed.

Yours sincerely,

Irene XY Wu

Xiangya School of Public Health

Central South University

Changsha, China

Comments to the Author:

1. Please ensure that you include a title page within your main document. We do appreciate that you have a title page document uploaded as a separate file; however, as per our author guidelines (http://journals.plos.org/plosone/s/submission-guidelines#loc-title-page) we do require this to be part of the manuscript file itself and not uploaded separately.

Could you therefore please include the title page into the beginning of your manuscript file itself, listing all authors and affiliations?

Response to comment:

Thank you for alerting us to this policy. We have added a tittle page to the main manuscript.

2. In your Data Availability statement, you have not specified where the minimal data set underlying the results described in your manuscript can be found. PLOS defines a study's minimal data set as the underlying data used to reach the conclusions drawn in the manuscript and any additional data required to replicate the reported study findings in their entirety. All PLOS journals require that the minimal data set be made fully available. For more information about our data policy, please see http://journals.plos.org/plosone/s/data-availability.Upon re-submitting your revised manuscript, please upload your study’s minimal underlying data set as either Supporting Information files or to a stable, public repository and include the relevant URLs, DOIs, or accession numbers within your revised cover letter. For a list of acceptable repositories, please see http://journals.plos.org/plosone/s/data-availability#loc-recommended-repositories. Any potentially identifying patient information must be fully anonymized.

Response to comment: 

Thank you for this comment. We have included all relevant data as Supporting information (S12).

Additional Editor comments:

This is a meta-analysis of HBV infection in Ghana. Given the high prevalence of HBV in West Africa, this is a reasonable study goal. The methods are well described and appropriate. The manuscript should be reviewed carefully by a native English speaker and/or a professional editing service.

Response to comment:

Thank you very much for your comments. We employed a professional proofreading service (Editage) to help in this regard.

Comment 1. It is unclear why only data from 2015 to 2019 were included? Certainly there are studies before that time period that are relevant.

Response to comment 1:

Thank you for your comment. The reason for conducting the study between 2015 and 2019 was informed by the fact that previous meta-analyses has synthesized evidence until 2015 [1]. Hence, the prevalence before 2015 was already known. The present systematic review aimed to summarize evidence from after 2015 which was not covered by previous systematic reviews. Also, given that in 2015 the World Health Organization set a target to eliminate hepatitis B by the year 2030 [1], this study sought to update the national prevalence, among other objectives, and aimed to track the HBV prevalence in Ghana for the 5 years following this declaration. We have added the aforementioned information to the manuscript to address these comments:

In 2015, the World Health Organization (WHO) set a target to eliminate hepatitis B by the year 2030 [1]. To explore and address any gaps in the available evidence and to track the HBV prevalence in Ghana following this declaration [1], we conducted a systematic review to compile updated information on the prevalence of HBV in Ghana from November 2015 – September 2019.

Comment 2: Line 2: Approximately 2 billion people have been exposed to HBV, but there certainly are not that many that are currently infected.

Response to comment 2:

Thank you for your comment. We agree that this is not the number currently infected; we therefore rephrased our statement as follows:

About 2 billion people worldwide are estimated to have been exposed to HBV, with almost one quarter of them having a chronic infection.

Comment 3: It would be helpful to mention and discuss briefly any other meta-analysis of HBV prevalence conducted in other Africa countries – particularly West Africa – for comparison.

Response to comment 3:

 Thank you very much for your comment; we do agree with you that related studies in parts of West Africa will enhance comparison. Results from a meta-analysis focused on the Nigerian population have been added and compared in the Discussion section. The related revision is as below:

In the adult population, overall HBV prevalence was found to be 8.36% in Ghana based on HBsAg seropositivity. This emphasizes that HBV remains a major public health concern in the country. In contrast, the current prevalence is significantly lower than the prevalence of 12.30% reported by earlier studies. However, a similar prevalence of 8.50% was reported quite recently in an epidemiological study in 2012. The results of this review are lower than the >8.80% reported in other parts of West Africa [2-4] and the prevalence for the African region at large [5]. For instance, a meta-analysis reported that the pooled prevalence of HBV in the Nigerian population is 13.6% [6]. A high prevalence was reported among different subgroups as well: 14.0% for blood donors, 14.1% for pregnant women attending antenatal clinics, 11.5% for children, and 14.0% for adults [6]. Nonetheless, the results from our study confirm the categorization of Ghana as a high HBV-endemic country (≥8.00%) which exceeds the reported global prevalence rate of 3.61 % [1, 7, 8].

Reviewers’ comments

Reviewer #1: 

Comment 1: The systematic review and meta-analysis done have not added any additional or new information to the prevalence of Hepatitis B in Ghana. This is due to limitations the authors themselves recognized. Studies do not cover the whole population, most studies are small and the number of studies in the heterogeneous populations few over the short period thus the basis for the meta analysis in some cases were two or three studies. The process thus looked more like an academic exercise rather than providing additional scientific knowledge. A previous meta-analysis covered 10 years: Ofori-Asenso R, Agyeman AA. Hepatitis B in Ghana: a systematic review & meta-analysis of prevalence studies (1995-2015). BMC Infect Dis. 2016 Mar 18;16:130. doi: 10.1186/s12879-016-1467-5. PMID: 26987556; PMCID: PMC4797341. Im not sure why the authors choose to do same over five years 2015-2019 instead of fifteen years, 1995 -2019

Response to comment 1: 

Thank you for your comments. As mentioned in our response to comment 1 from the Editor, the reason for conducting the study between 2015 and 2019 was informed by the fact that meta-analyses have synthesized evidence until 2015 [1]. Hence, the prevalence before 2015 was already known. The present systematic review aimed to summarize evidence from after 2015 which was not covered by previous systematic reviews. Also, as the World Health Organization set a target in 2015 to eliminate hepatitis B by the year 2030 [1], this study sought to update the national prevalence, amongst other objectives, and aimed to track HBV prevalence in Ghana for the 5 years following this declaration.

Furthermore, this study revealed new findings about hepatitis B infection in the Ghanaian population. This included the prevalence rate of hepatitis among preschool children which was not revealed by the two previous meta-analyses since 1965, Schweitzer et al. [5] and Ofori-Asenso and Agyeman [9]. The above studies also stated the limitations in obtaining a nationwide prevalence rate. Schweitzer included only three studies from Ghana and used them to approximate the national prevalence. Additionally, Ofori-Asenso and Agyeman could not find any related studies for four regions of the country and acknowledged that the results could not be said to be the national prevalence. To address this comment, we made revisions in the Introduction and Discussion sections:

Introduction:

In 2015, the World Health Organization (WHO) set a target to eliminate hepatitis B by the year 2030 [1]. To explore and address any gaps in the available evidence and to track the HBV prevalence in Ghana following this declaration [1], we conducted a systematic review to compile updated information on the prevalence of HBV in Ghana from November 2015 – September 2019.

Discussion:

There are several limitations of this review. Although we comprehensively searched the literature, an overall national prevalence of HBV could not be determined because we were not able to identify any related publications for the Upper West Region. The prevalence determined in this study was therefore based on subpopulations and hence cannot be said to fully represent the general Ghanaian population. Furthermore, most of the studies (13/21, 62%) were conducted in urban settings and in adult populations (18/21, 86%). Hence, the prevalence is more representative of the adult, urban population. The aforementioned information also revealed the research priorities for future studies: HBV prevalence among children and adolescents, in populations from the Upper West region of Ghana, and in rural areas.

Comment 2: Results-There was no PRISMA diagram. There are no random effect plots

Response to comment 2: 

Thank you for your comments. According to the submission guidelines, figures should be uploaded separately. We uploaded the PRISMA diagram as Fig. 1, so it was not embedded in the results. Several random effects plots were uploaded (e.g. Fig. 2 and the supporting figures for all subgroups). They are presented after the main text in the generated PDF file.

Comment 3: Discussion- This needed to have more depth. Ghana started a national expanded programme on Hepatitis immunization in 2002. The high prevalence in adolescents in contrast to children and adults to be put in context. Also the quality or generation of HBsAg tests used should have been analyzed in the Meta analysis to enrich the discussions.

Response to comment 3: 

Thank you very much for your comments; it is an important observation that you made. We expected that after the national program of immunization, a reduction in prevalence should be seen among adolescence. However, it is observed that as age increases, the antibody titer decreases. Because there were not sufficient studies to explore this, we propose that the high prevalence of hepatitis B among adolescents was also related to adolescents’ risky behavior and parents not openly discussing sexual behavior with teenagers. We linked to this idea in the Discussion section:

It was noted that Ghana started a national expanded program for hepatitis immunization in 2002 [10], which should have reduced the prevalence of HBV infections, especially among adolescents. However, an alarming prevalence of about 14.30% was identified among adolescents in this study. There were not sufficient studies to explore this; however, adolescence is characterized by youthful exuberance, sexual experimentation, body modifications (tattooing, piercing, etc.), and risky sexual behaviors, which are all risk factors for HBV transmission [11]. Furthermore, in Ghana, adolescents do not discuss sex openly with their parents; they resort to friends who are as well inexperienced with sexually related matters and may end up engaged in risky behaviors. This suggests the need for further research to fully explore the cause of the high HBV prevalence in this population, regardless of the national immunization program.

We agree with you that the quality and generation of HBsAg tests should have been analyzed in the meta-analysis to enrich the Discussion. We analyzed the quality of tests in the subgroup analysis (Table 1), which indicates that ELISAs were more accurate in detecting HBV compared to RDT. However, RDT is the more commonly used test in Ghana’s blood donation centers. This certainly attracted our attention, and was emphasized in the Discussion as follows:

Furthermore, RDTs are the most commonly used kits in developing countries because of their ease of use and lower cost. Meanwhile, Mutocheluh and colleagues examined five brands of these kits on the Ghanaian market, finding a sensitivity lower than 60% and raising serious concerns when used in blood donation centers to exclude HBV [12]. The present study emphasizes the need for robust screening techniques and uncompromising adherence to the transfusion policy in Ghana which advocates for the need to screen for transfusion-transmissible diseases such as HIV, HBV/HCV, and syphilis.

Reviewer #2: ACCEPT with some good grammatical revisions please

This is a well-conducted and useful meta-analysis. The methods in the literature search and statistical approaches are appropriate and sound. The subgroup analyses were very revealing in terms of pockets of high-need populations within Ghana. These were clearly linked to explanations within the discussion that describe attributes which predispose to infection and/or increased morbidity (i.e. adolescent sexual practices; HBV detection kids with poor sensitivity in the blood donor population; etc.). 

The pooled prevalence itself is sufficient to raise alarm bells, but the extremely high rates in certain populations (adolescents, pregnant women, HIV+) suggest additional need for targeted intervention and care.

Response to comment: 

Thank you very much for your comments. We employed a professional proofreading service (Editage) to help in this regard.

References:

1. Guidelines for the prevention, care and treatment of persons with chronic hepatitis B infection. Geneva: World Health Organization; 2015.

2. Diarra B, Yonli AT, Ouattara AK, Zohoncon TM, Traore L, Nadembega C, et al. World hepatitis day in Burkina Faso, 2017: seroprevalence and vaccination against hepatitis B virus to achieve the 2030 elimination goal. Virol J. 2018;15: 121. doi: 10.1186/s12985-018-1032-5. 2018.

3. Nejo Y, Faneye AO, Olusola B, Bakarey S, Olayinka A, Motayo B, et al. Hepatitis B virus infection among sexually active individuals in Nigeria: a cross-sectional study. Pan Afr Med J. 2018;30: 155. doi: 10.11604/pamj.2018.30.155.14886.

4. Adesina OA, Akinyemi JO, Ogunbosi BO, Michael OS, Awolude OA, Adewole IF. Seroprevalence and factors associated with hepatitis C coinfection among HIV‑ positive pregnant women at the University College Hospital, Ibadan, Nigeria. Trop J Obstet Gynaecol. 2016;33: 153–158. doi: 10.4103/0189-5117.192216.

5. Schweitzer A, Horn J, Mikolajczyk RT, Krause G, Ott JJ. Estimations of worldwide prevalence of chronic hepatitis B virus infection: a systematic review of data published between 1965 and 2013. Lancet. 2015;386: 1546–55. doi: 10.1016/S0140-6736(15)61412-X.

6. Musa BM, Bussell S, Borodo MM, Samaila AA, Femi OL. Prevalence of hepatitis B virus infection in Nigeria, 2000-2013: A systematic review and meta-analysis. Niger J Clin Pract. 2015;18: 163–172. doi: 10.4103/1119-3077.151035. ( ).

7. Global Hepatitis Report 2017. Geneva: World Health Organization; 2017.

8. Spearman WC, Afihene M, Ally R, Apica B, Awuku Y, Cunha, L, et al. Hepatitis B in sub-Saharan Africa: strategies to achieve the 2030 elimination targets. Lancet Gastroenterol Hepatol. 2017;2: 900–909. doi: 10.1016/S2468-1253(17)30295-9.

9. Ofori-Asenso R, Agyeman AA. Hepatitis B in Ghana: a systematic review & meta-analysis of prevalence studies (1995-2015). BMC Infect Dis. 2016; 16: 130. doi: 10.1186/s12879-016-1467-5. .

10. Ghana Health Service 2016 Annual Report. Accra: Ghana Health Service; 2017.

11. Yang S, Wang D, Zhang Y, Yu C, Ren J, Xu K, et al. Transmission of hepatitis B and C virus infection through body piercing: a systematic review and meta-analysis. Medicine (Baltimore). 2015;94: e1893. doi: 10.1097/MD.0000000000001893.

12. Mohamed Mutocheluh, M.O., Theophilus B Kwofie, Tahiru Akadigo, Emmanuel Appau and a.P.W. Narkwa, Risk factors associated with hepatitis B exposureand the reliability of five rapid kits commonly used for screening blood donors in Ghana. BMC Research Notes 2014. 7: p. 873.

---

## [Decision Letter · Decision Letter 1]

26 May 2020

Prevalence of viral hepatitis B in Ghana between 2015 and 2019: a systematic review and meta-analysis

PONE-D-20-02736R1

Dear Dr. Wu,

We are pleased to inform you that your manuscript has been judged scientifically suitable for publication and will be formally accepted for publication once it complies with all outstanding technical requirements.

With kind regards,

Jason Blackard, PhD

Academic Editor

PLOS ONE

Additional Editor Comments (optional):

None

Reviewers' comments:

Reviewer's Responses to Questions

**Comments to the Author**

1. If the authors have adequately addressed your comments raised in a previous round of review and you feel that this manuscript is now acceptable for publication, you may indicate that here to bypass the “Comments to the Author” section, enter your conflict of interest statement in the “Confidential to Editor” section, and submit your "Accept" recommendation.

Reviewer #1: All comments have been addressed

2. Is the manuscript technically sound, and do the data support the conclusions?

Reviewer #1: (No Response)

3. Has the statistical analysis been performed appropriately and rigorously? 

Reviewer #1: (No Response)

4. Have the authors made all data underlying the findings in their manuscript fully available?

Reviewer #1: (No Response)

5. Is the manuscript presented in an intelligible fashion and written in standard English?

Reviewer #1: (No Response)

6. Review Comments to the Author

Reviewer #1: (No Response)

7. PLOS authors have the option to publish the peer review history of their article (what does this mean?). If published, this will include your full peer review and any attached files.

Reviewer #1: No

---

## [Editor Report · Acceptance letter]

29 May 2020

PONE-D-20-02736R1 

Prevalence of viral hepatitis B in Ghana between 2015 and 2019: a systematic review and meta-analysis 

Dear Dr. Wu:

I am pleased to inform you that your manuscript has been deemed suitable for publication in PLOS ONE. Congratulations! Your manuscript is now with our production department. 

With kind regards,

on behalf of

Dr. Jason Blackard 

Academic Editor

PLOS ONE